# Diabetes Mellitus and Hearing Loss: A Complex Relationship

**DOI:** 10.3390/medicina59020269

**Published:** 2023-01-30

**Authors:** Federico Maria Gioacchini, Davide Pisani, Pasquale Viola, Alessia Astorina, Alfonso Scarpa, Fernanda Asprella Libonati, Michele Tulli, Massimo Re, Giuseppe Chiarella

**Affiliations:** 1ENT Unit, Department of Clinical and Molecular Sciences, Polytechnic University of Marche, 60121 Ancona, Italy; 2Unit of Audiology, Regional Centre of Cochlear Implants and ENT Diseases, Department of Experimental and Clinical Medicine, Magna Graecia University, 88100 Catanzaro, Italy; 3Department of Medicine and Surgery, University of Salerno, 84084 Fisciano, Italy; 4Indipendent Researcher, 75100 Matera, Italy

**Keywords:** diabetes mellitus, hearing loss, sensorineural hearing loss

## Abstract

*Background and Objectives*: Discussion is open about the relationship between diabetes (DM) and hearing loss (HL). There is a lot of evidence in the literature suggesting a causal link between these conditions, beyond being considered simple comorbidities. The difficulty in identifying populations free from confounding factors makes it difficult to reach definitive conclusions on the pathophysiological mechanisms at play. Nonetheless, there is numerous evidence that demonstrates how the population affected by DM is more affected by sensorineural HL (SNHL) and exhibit a higher prevalence of idiopathic sudden sensorineural HL (ISSNHL). *Materials and Methods*: Articles reporting potentially relevant information were reviewed, and the most significant results are discussed in this article. Starting from the possible mechanisms relating to auditory impairment in the diabetic condition, this article summarizes the studies on auditory evaluation in subjects with DM1 and DM2 and addresses the relationship between DM and ISSNHL. *Results*: DM is considered a risk factor for SNHL, although some studies have reported no relationship when the associations were adjusted for age, gender, and hypertension. Macro and microvascular insults that cause decreased blood flow, oxygen exchange, and ion transport are major complications of hypertension and DM and can have a direct effect on the sensory and support cells of the cochlea. *Conclusions*: Given the difficulty of carrying out studies on populations without confounding factors, new laboratory studies are strongly required to clarify which specific physiopathological mechanisms underlie the diabetic damage caused to the hearing organs and how pharmacological management may contribute to counteracting the pathophysiological effects of the diabetic condition on the auditory system.

## 1. Introduction

There is growing attention paid to the relationship between diabetes and inner ear structures. [1,2,3,4,5,6,7,8,9,10,11,12,13,14,15,16,17,18,19,20,21,22,23,24,25] In particular, there is a wealth of evidence suggesting that hearing loss (HL) may be a disabling complication of diabetes mellitus (DM) [26,27]. The consequences of DM affecting the ear can include the ability to understand speech, the risk of falling, and the onset of tinnitus, with concerns ranging from a lowered quality of life to an increase in mortality. Although studies of DM and HL often show an association, the interpretation of causality is often difficult. Few studies examined the pathological interaction between DM and HL due to the absence of an appropriate animal model of DM and the lack of access to cochlear tissue in humans in vivo [28]. Studies on the possible relationship between DM and HL are often biased by noise exposure, ototoxic drug use, and confounding factors such as age, gender, duration of DM, glycemic control, smoking, and other metabolic and cardiovascular comorbidities [29]. DM is a chronic multisystem condition characterized by high levels of glucose in the blood and urine due to the inadequate production or use of insulin. The estimated worldwide prevalence of DM (types 1 and 2) is 6.4%, with an expected prevalence of 7.7% by 2030 [30]. Type 1 DM (DM1), also known as autoimmune diabetes, is characterized by insulin deficiency due to the loss of pancreatic β-cells and accounts for less than 10% of diabetes cases. Type 2 DM (DM2) results from the progressive loss of β-cell insulin secretion and an inadequate response to insulin secretion and accounts for approximately 90–95% of DM cases [31]. DM is a major cause of death and is associated with numerous comorbidities. Micro and macrovascular lesions can cause retinopathy, peripheral neuropathy, and nephropathy. Many patients also have sensorineural HL (SNHL). Moreover, despite HL having been identified as a possible consequence of DM, hearing assessment is not included in the most recent protocol for the assessment of DM comorbidities [32]. It is believed that high blood glucose levels can damage the vessels of the stria vascularis and nerves, affecting hearing [33]. This article explores and discusses the complex relationship between DM and HL.

## 2. Materials and Methods

This study was principally designed to review the most recent evidence related to DM and auditory system impairment. In October 2022, a literature search was performed using the following string on PubMed: (“Diabetes Mellitus, Type 2” [Mesh] OR “Diabetes Mellitus, Type 1” [Mesh] OR “Diabetes Mellitus” [Mesh]) AND (“Hearing Loss” [Mesh] OR “Hearing Loss, Unilateral” [Mesh] OR “Hearing Loss, Sensorineural” [Mesh] OR “Hearing Loss, Sudden” [Mesh]). Only articles published in the English language were considered. The abstracts and titles obtained were screened independently by two of the authors (F.M.G. and P.V.), who subsequently discussed disagreements of citation inclusion. The articles reporting potentially relevant information were read in full. The files were then revised, and the most significant results were discussed in this article.

## 3. Results

### 3.1. Anatomy and Physiology of Cochlear Organ

The cochlea is the mammalian organ of hearing. It is a coiled, fluid-filled tube that is composed of three compartments known as the scala vestibuli, scala media, and scala tympani which run along its length [34]. Corti’s organ is the hearing organ within the cochlea, and it includes hair and supporting cells. Two types of hair cells are present, inner and outer, with inner hair cells (IHCs) being the true sensory cell type, sending impulses via the auditory nerve, and outer hair cells (OHCs) enhancing cochlear performance by increasing selectivity and sensitivity [35]. Although they are not the actual sound transducers, supporting cells are essential, and without them, hearing would not be possible. There are approximately 15 supporting cells per each IHC, and the different types of supporting cells include: border, inner phalangeal, pillar, Deiters’ (outer phalangeal cells), and Hensen’s cells [36]. The basilar membrane (BM) is an extracellular matrix that separates the scala media from the scala tympani and vibrates in response to sound-induced pressure changes in the cochlear fluids. Movements of the BM are, in turn, detected by the hair cells in the organ of Corti. The tectorial membrane is a ribbon-like strip of extracellular matrix that spirals along the length of the organ of Corti. It is attached along its medial edge to the surface of the spiral limbus, extends across the internal spiral sulcus, and lies over the sensory epithelium, attaching along its lower lateral surface to the tips of the hair bundles of the OHCs [34]. The stria vascularis (SV) is the highly vascularized area located in the lateral wall of the cochlea that contributes to cochlear homeostasis in two ways. First of all, the SV directly consents to maintain the endocochlear potential, which is required for the process of hair cell transduction to occur. This action is achieved by mean ion transport proteins that perform active potassium ion (K+) recycling between the endolymph of the scala media and the perilymph of the scala tympani. In addition, the SV contains the cochlear blood–labyrinth barrier, which tightly controls the transfer of material from the SV capillary network into the endolymph [37].

### 3.2. Possible Mechanisms for Auditory Impairment in Diabetic Condition

Although several theories have been proposed (e.g., microangiopathy, advanced glycation end products, reactive oxidative stress and mitochondrial dysfunction, demyelination of the auditory nerve, spiral ganglion loss, and atrophic changes of organ of Corti cells), the exact underlying mechanisms responsible for diabetes-induced damage to the auditory system remain uncertain. Many studies performed on animal or human models tried to investigate the alterations occurring in cochlear substructures after a prolonged diabetic state (Table 1). Smith et al., demonstrated basement-membrane thickening in the SV capillaries of animals with a 6-month duration of insulin-dependent DM [38]. Tsuda et al., analyzed the hearing function and histopathologic changes in the cochlea of a type 2 diabetes model mouse in comparison to control mice. They observed a significantly elevated auditory brainstem response (ABR) threshold at 8–10 months. In addition, a histopathologic evaluation showed narrower capillary lumens in the SV of type 2 DM mice in comparison to normal controls. Another important observation is connected to the capillary density in SV that was decreased in type 2 DM mice at 9 and 17 months of age [39]. Lee et al., analyzed the effect of type 1 DM in Akita mouse models in comparison to wild-type mice. The Akita group showed significantly decreased thickness of the SV in the middle and basal turns. Animal models of type 1 DM also showed loss of cells and damage to mitochondria in the spiral ganglion neurons compared to wild-type mice. Moreover, in the Akita group, the expression of Na+/K+-ATPase α-1 in type II and IV fibrocytes, and the SV in the middle to basal turn, was decreased, in comparison to the wild-type group [40]. A similar result was already observed by Meyer zum Gottesberge et al., who analyzed Zucker Diabetic Fatty rats as animal models of type 2 DM. The authors found that Na+/K+-ATPase was altered under diabetic conditions in Zucker Diabetic Fatty rats and the ß1-subunit was significantly reduced in the type II fibrocytes compared to normoglycemic conditions [41]. In a recent study on a mouse model of DM, Lyu et al., found disruptions of the mitochondrial structure and morphology in the SV [42]. Analogous results were observed for the mitochondria of IHC, OHC, and synapses of spiral ganglion neurons when compared to wild-type controls. Raynor et al., also found that rats with metabolically generated DM showed a significant decrease in neuroganglion cell density in the basal turn of the spiral ganglion [43]. Other types of studies focused on the insulin resistance, or hyperinsulinemia, that characterizes type 2 DM and speculated about the possibility of auditory damage directly induced by insulin level alterations. This theory may be supported by a study performed by Huerzeler et al., who showed expression of the insulin receptor within the Corti organ [44]. Based on this finding, Palbrink et al., investigated the role of insulin signaling and insulin resistance using high-fat-diet mice. The size of the inner ear endolymphatic fluid compartment was measured after 30 days using MRI and gadolinium contrast. The authors observed an expansion of the endolymphatic fluid compartment in high-fat-diet mice in comparison to mice fed a control diet, with hydrops developing in the right as well as in the left ears [45]. Concerning investigations performed with human models, Fukushima et al. [46] analyzed temporal bones from 18 patients with type 2 DM and 26 age-matched controls. Morphometric measurements of vessel wall thickness in the BM and SV were performed, with results showing significantly thicker vessel walls in the diabetic specimen than in those in the controls. Furthermore, they found atrophy of SV in most cochlear turns of the diabetic group and a loss of cochlear OHC that was significantly greater than in the controls. On the other hand, there was no significant difference in the number of spiral ganglion cells or IHCs between groups. In addition to specific cochlear changes, auditory nerve demyelination is also suspected to promote acoustic deterioration in patients affected by chronically high glycemic levels. The best way to determine neurologic transmission reduction is by testing the ABR, which assesses retrocochlear neural pathways. The largest study to date was performed by Vaughan et al., and assessed 791 subjects. The results showed that patients with diabetes had significantly delayed latencies of ABR wave III and V in the right ear and significantly prolonged interpeak I–III and I–V latencies in both ears [47].

#### 3.2.1. Auditory Evaluation in Subjects with DM1

DM1 is mainly due to a lack of insulin caused by the destruction of pancreatic β-cells by an autoimmune process. DM1 differs from type 2 in etiology, pathogenesis, clinical features, and predisposition to complications. DM1 appears at earlier ages than DM2, and therefore, its complications are already present in young adulthood. In recent years, two systematic reviews with meta-analyses investigated the audiological modifications in subjects affected by DM1. In 2017, Teng et al., tried to establish the effective relationship between the presence of DM1 and auditory dysfunction. Fifteen studies were included, and the authors observed a significant relationship between the presence of DM1 and an increased risk for developing mild and subclinical hearing impairment [48]. Likewise, in 2018, Mujica-Mota et al., considering 21 papers, found that DM1 patients have a significantly greater prevalence of SNHL compared to controls. Hearing assessment was performed by mean Pure Tone Audiometry (PTA) in most of the selected studies, and auditory function was only explored in some reports with otoacoustic emissions (OAE) or ABR. Actually, in consideration of the young age of many DM1 patients, the HL may not be present at PTA during the early pathological phase. For this reason, some authors decided to investigate the presence of an initial cochlear alteration using transient evoked OAE (TEOAE) and distortion product OAE (DPOAE) in normal-hearing young adults with DM1, demonstrating a decreased OAE response in DM1 subjects, suggesting its damaging effects on the cochlea [49].

#### 3.2.2. Auditory Evaluation in Subjects with DM2

Akinpelu et al., included 18 studies in a meta-analysis about the possible association between DM2 and hearing alterations. The authors concluded that DM2 patients have significantly higher incidence of mild HL when compared with controls. DM2 patients had poorer hearing at all frequencies, with greater impairment at 6 and 8 kHz. Interestingly, these patients had a threefold delay in ABR wave V latency, generated from the inferior colliculus that is the principal brainstem nucleus of the auditory pathway. These data also suggest retrocochlear involvement. Moreover, the incidence of HL was higher for older diabetics. Control patients also showed an increase in the incidence of HL with increasing age; however, the increase was greater among the diabetic group [50]. Some recent reports corroborated these findings, showing a higher prevalence of HL in DM2. In 2017, Ren et al., demonstrated that mild HL is a common condition in DM2, with high-frequency involvement. In this population study, DM patients with HL were older and males were more affected compared to DM patients with normal hearing thresholds. The mean PTA thresholds were greater in DM patients than in controls for all frequencies, mainly at high frequencies; hearing damage was mild for pure high-frequency HL, becoming greater when progressing to low-frequency involvement. Other results that can be considered interesting and point towards a neuropathic involvement are the Michigan Neuropathy Screening Instrument (MNSI), Semmes–Weinstein Monofilament (SWM), and vibration perception threshold (VPT) scores (used to evaluate diabetic neuropathy, including both large and small nerve fiber lesions) in DM patients that were independent risk factors for HL. [51]. In 2018, Gupta et al., performed a longitudinal study with 139,909 women to examine the relationship between DM2 and self-reported HL. Based on administered questionnaires, HL was reported as moderate or worse (categorized as a ‘moderate or severe’ hearing problem or ‘moderate hearing trouble or deaf’). Compared with normal controls, women with DM2 were at higher risk for moderate or worse HL; moreover, the study demonstrated a higher risk of moderate or worse HL in DM2 subjects lasting for 8 or more years compared with individuals without DM2. These authors carefully adjusted for potentially important confounders, finding that the increased risk of HL in DM2 is independent from BMI and age [52]. Recently, Al-Rubeaan et al., analyzed 157 patients with DM2. The authors decided to limit the age window (30–60 years) to eliminate the effects of natural aging on hearing. The results showed a higher frequency of HL in patients with glycated hemoglobin levels ≥8%. In the multivariate logistic regression analysis, the most important factors associated with HL were longer diabetes duration, poor glycemic control, and the presence of hypertension. Likewise, for DM1, an important drawback of the standard diagnostic process is that some patients with DM2 may show normal hearing thresholds by conventional PTA and ABR. Therefore, it could be necessary use some different tools to detect early hearing impairment [53]. In 2020, Li et al., demonstrated that extended high-frequency audiometry (HFA) and DPOAE might help to identify initial cochlear dysfunction and high-frequency HL in DM patients with normal hearing thresholds of conventional frequencies. The evaluation of hearing threshold by conventional PTA and HFA documents the IHC and OHC function and related auditory pathways. On the other hand, the DPOAE investigates the cochlear functional state, mainly the outer hair cells. This study demonstrated that the combination of these two tests, one subjective and the other objective, is more sensitive in detecting the inner ear damage of DM patients at the earlier stage than conventional PTA [54].

### 3.3. Diabetes and Idiopathic Sudden Sensorineural Hearing Loss (ISSNHL)

Only a few studies investigated the possible connections between diabetes and ISSNHL, which is defined as a sudden onset of unilateral sensorineural HL of 30 dB or more over at least three contiguous audiometric frequencies [55,56,57]. Weng et al., reported the clinical features of 67 patients with diabetes and ISSNHL. Profound HL was very common in this cohort of patients (44.8%). Glucocorticoids were administered for therapy, prednisolone 1 mg/kg per day for at least 7 days. In a follow-up, the low- and middle-tone hearing thresholds in the involved ears showed greater improvement than at high frequencies. Hearing consistently improved within 2 months after disease onset, but improvements were rare thereafter [58]. Lin et al., performed a retrospective cohort study analyzing the differences among DM and non-DM patients in consideration of the hearing levels achieved after steroid treatment for ISSNHL. The mean pre-treatment PTA (affected ear) was 71.4 + 24.0 dB; range 31.3 to 110.0 dB, while the mean post-treatment PTA was 51.76 + 29.6 dB; range 12.5 to 110.0 dB. Regression analyses adjusted for gender, age, pre-treatment hearing, treatment delay time, and all comorbidities showed that the probability of major improvement in the PTA was significantly higher in patients without diabetes compared to those with diabetes [59]. Ryu et al., assuming that microvascular damages in hyperglycemic patients would negatively affect the prognosis of ISSHL, analyzed the prognostic value of hyperglycemia and DM in predicting hearing recovery after ISSNHL. To avoid the misclassification of the population studied, the authors classified patients according to history, oral glucose tolerance test, and the hemoglobin A1c level. Among 94 patients, 45 were classified into the normal glucose tolerance group, 28 into the pre-DM group, and 21 into the DM group. The hearing recovery rate of the normal glucose tolerance (normoglycemia) group was significantly better than the impaired glucose regulation group (pre-DM and DM) [60]. In contrast, Seo et al., in their retrospective study evaluating DM as a prognostic factor in ISSNHL, concluded that DM itself may not influence the ISSNHL prognosis. Based on their analysis of 403 patients, the authors noted that DM patients (*n* = 94, 23.3%) usually presented with severe HL and required longer hospitalization than subjects with normal glucose levels. The DM patients had a higher initial hearing threshold, more hypertension, and less frequent hearing recovery than those without diabetes. However, when age, sex, and initial hearing level were adjusted by propensity score matching, the DM patients and matched controls yielded similar treatment results [61].

## 4. Discussion

DM is also considered a risk factor for SNHL, although some studies have reported no relationship when the associations were adjusted for age, gender, and hypertension. Macro and microvascular insults that cause decreased blood flow, alterations in oxygen exchange, and ion transport are major complications of hypertension and DM and can have a direct effect on the sensory and support cells of the cochlea. A balanced diet and healthy lifestyle can prevent the progression of DM and SNHL; if necessary, vasodilators can be used to improve nerve blood flow.

A large majority of published reports investigating hearing impairment in DM confirm the presence of a mild HL for high frequencies, suggesting that the most damaged inner ear structures are those placed on the cochlear basal turn that seems to be a “fragile” inner ear portion, potentially affected by different pathological events. In fact, other audiological conditions also represent important confounding factors during the clinical evaluation of diabetic HL. The first consideration must be age-related hearing loss (ARHL), which often starts with worsening high-frequency sound perception. Furthermore, noise-induced hearing loss (NIHL) is another condition that leads to selective damage in high-frequency sound transduction.

For this reason, new laboratory studies are strongly required to clarify which are the specific physiopathological mechanisms that promote diabetic damage towards the hearing organs. These studies should also try to define if the auditory diabetic damage shares the same molecular mechanisms as those that are associated with the aging process and chronic noise exposure. Clarifying whether these mechanisms have common pathways can help identify shared therapeutic approaches.

Furthermore, as regards future clinical trials, it would be necessary to standardize a series of hearing tests with the aim of homogeneously examining the diabetic patients’ auditory pathway, as is already performed during oculistic and nephrological evaluations. Our review clarified how some tests can be used as a screening procedure to identify hearing impairments that have already engendered clinical consequences, such as pure-tone and speech audiometry. Meanwhile, other, more sophisticated hearing tests may be useful for detecting subclinical cochlear damage. Well-designed longitudinal studies are also strongly recommended to investigate if the progression of auditory alterations differs on the basis of diabetes status/duration, as well as to ascertain if different pharmacological management may contribute to counteracting the pathophysiological impact of the diabetic condition on the auditory system.

HL can strongly compromise the quality of life in diabetic subjects, and the inclusion of hearing evaluation in the American Diabetes Association recommended comprehensive medical assessment would be useful. Interventions aimed at controlling factors that may cause morphological and functional changes in the cochlea are critical in managing diabetic hearing damage.

## 5. Conclusions

Individuals with DM are at increased risk of hearing impairment, although this particular population may frequently have confounding comorbidities. Since the severity, course, and consequences of hearing damage can be influenced by the medical treatment of DM and by the medical and rehabilitative management of HL, we believe it is important to include a detailed assessment of the hearing threshold in the diagnostic work-up of diabetic patients.

Interventions aimed at controlling factors that may cause morphological and functional changes in the cochlea are critical in managing diabetic hearing damage.

## Figures and Tables

**Table 1 medicina-59-00269-t001:** Inner ear alterations observed in experimental models with (Diabetes Mellitus) DM.

Authors	Year	Inner Ear Alterations
Smith et al. [38]	1995	microangiopathy with capillary basement membrane thickening
Raynor et al. [43]	1995	decrease in the neuroganglion cell density in the basal turn of the spiral ganglion
Fukushima et al. [46]	2006	atrophy of stria vascularis in most cochlear turns
		loss of cochlear outer hair cells
Meyer zum Gottesberge [41]	2015	downregulation of Na+/K+-ATPase pump
Tsuda et al. [39]	2016	capillary lumens in the cochlea stria vascularis were narrower
		capillary density in the stria vascularis was decreased
Lee et al. [40]	2020	decreased thickness of the stria vascularis in the cochlear middle and basal turns
		loss of cells and damage to mitochondria in the spiral ganglion neurons
		the expression of Na+/K+-ATPase α-1 in type II and IV fibrocytes was decreased
		the stria vascularis in the middle and basal turn was decreased
Palbrink et al. [45]	2020	expansion of the endolymphatic fluid compartment and hydrops
Lyu et al. [42]	2021	disruption of the mitochondrial structure in the stria vascularis
		disruption of the mitochondria structure of inner and outer hair cells
		disruption of the mitochondrial structure in synapses of spiral ganglion neurons

## Data Availability

Not applicable.

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
