# Peer review of "Diabetes Mellitus and Hearing Loss: A Complex Relationship"

_medicina, 2023, doi:10.3390/medicina59020269_

Round 1
Reviewer 1 Report
The manuscript reviews the relation between diabetes (DM) and sensorineural hearing loss (HL). The review includes information about cochlear anatomy, and includes both animal and human studies. The association between DM and HL is reported. As noted, patients with DM might have HL from other causes, such as noise exposure of aging. For that reason, it cannot be said definitively that HL is a direct consequence of DM. Overall, the comprehensive review is a strength of the manuscript. But the range of topics means that details must sometime be omitted; for example, some readers might have wanted a bit more information about the stria vascularis, since that may be the peripheral structure most likely to be affected by DM. There are a few places where the text could be made clearer, as noted in the Specific Comments.
Specific comments:
Line 80: I don’t think the word “qualitative” is correct here. Both of the listed effects of OHC activity are measurable (i.e., quantifiable) as changes in the responses of primary auditory neurons.
Line 87: “Motions” in the fluid is not quite accurate; I suggest “pressure changes” or “pressure waves” or “pressure gradients” as possible alternatives.
Line 90: “Stretches” can be misleading, as it could imply that the membrane is under tension (it isn’t); “extends” is a possible replacement.
Lines 179-173: What was the outcome of these OAE studies?
Line 213: The sentence mentions the “probability of improvement”, but says nothing about the magnitude of improvement (and for that matter, the magnitude of HL prior to improvement)? The review would be more informative if numbers were provided, here and elsewhere.
Reviewer 2 Report
Dear editors/ authors
I find as an ENT doctor very satisfying to see a review of hearing loss in an Endocrinology article. The part where the physiopatology is described is very thorough. I missed seeing mention to a few inner ear diseases (which would cause hearing loss, dizziness and tinnitus) that have relations to DM and glycemic issues though, like endolymphatic Hydrops… but maybe that would be too specific.
This article is very important because it brings attention to hearing impairments and inner ear function in diabetic patients, which should be one of the concerns of any doctor treating theses patients.
Sincerely
